# Exploring the Phytochemical Profiles and Antioxidant, Antidiabetic, and Antihemolytic Properties of *Sauropus androgynus* Dried Leaf Extracts for Ruminant Health and Production

**DOI:** 10.3390/molecules27238580

**Published:** 2022-12-05

**Authors:** Rayudika Aprilia Patindra Purba, Pramote Paengkoum

**Affiliations:** Institute of Agricultural Technology, School of Animal Technology and Innovation, Suranaree University of Technology, Nakhon Ratchasima 30000, Thailand

**Keywords:** HPLC, medicinal plants, plant extracts, phytochemical profiles, scavenging, ruminant health

## Abstract

*Sauropus androgynus* has become an essential plant in pharmaceutical formulations due to its beneficial antioxidant phytochemical components, participating in the antioxidant defense system and playing an important role in protecting human health. However, no research has been conducted on ruminant animals. This study aimed to evaluate the phytochemical profiles and biological potential of *S. androgynus* leaf extracts for ruminant health. Methanolic and hexanoic extracts from each commercially and noncommercially cultivated site were prepared over the course of five consecutive months. By means of HPLC-DAD, vitamins (ascorbic acid), essential oils (eugenol), tannins (gallic acid), cinnamic acids (caffeic acid, syringic acid, *p*-coumaric acid, sinapic acid and ferulic acid), and flavonoids (catechin, rutin, myricetin, quercetin, apigenin, and kaempferol) were detected. Variations in phytochemical composition were depending on solvent type but not on cultivation site or sample period. Methanolic extracts contained more phytochemicals than hexanoic extracts. Ascorbic acid and rutin were discovered to be the two most abundant phytochemicals in the methanolic extract of *S. androgynus* leaf, followed by essential oils, cinnamic acids, and tannins. In comparison to hexanoic extract, methanolic extract of *S. androgynus* demonstrated to be more efficient against oxidation scavenging: 1,1-diphenyl-2-picrylhydrazyl (IC_50_ = 13.14 ± 0.055 (mg/mL)), nitric oxide (IC_50_ = 55.02 ± 1.338 (mg/mL)) and superoxide (IC_50_ = 25.31 ± 0.886 (mg/mL)), as well as α-glucosidase inhibitory activity (IC_50_ = 9.83 ± 0.032 (mg/mL)). Similarly, methanolic was found to be more protective than hexanoic against oxidative damage in ruminant erythrocytes, with IC_50_ values (mg/mL) for hemoglobin oxidation, lipid peroxidation, and hemolysis of 11.96 ± 0.011, 13.54 ± 0.012, and 5.940 ± 0.005, respectively. These findings suggested that the leaves of *S. androgynus* are a prospective source of phytochemical substances with health-promoting qualities for ruminant production.

## 1. Introduction

In recent years, phytochemicals generated from plants such as herbs, e.g., natural phenolic compounds and vitamin C, have attracted interest due to the health-improving features that they possess. One of these properties is the potential for phytochemicals to serve as a viable alternative to natural antioxidants in the mitigation of oxidative stress [1,2]. Oxidative stress is brought on by the existence of reactive oxygen species (ROS) within the cell, which are able to triumph over the cell’s own inherent antioxidant defenses. In ruminants, oxidative stress may be involved in a variety of pathological disorders, including those related to animal production and general wellbeing [3]. Because there are often no outward signs of oxidative stress, its diagnosis requires the use of specialized analytical techniques. Traditionally, a single measurement of total antioxidant capacity provides informative data, such as the 1-diphenyl-2-picrylhydrazyl (DPPH) radical scavenging assay [4], which adequately explains the dynamic balance of pro-oxidants and antioxidants in the blood. In ruminant medicine, however, the field of oxidative stress is still in its infancy. Despite the fact that oxidative stress has been linked to a variety of illnesses, there is still much to learn about its role in ruminant health and productivity, such as the impact of natural phenolic compounds and vitamin C, particularly from medicinal plants or herbs [2,5].

*Sauropus androgynus* L. Merr (Phyllanthaceae), a perennial shrub sometimes known as a multigreen plant, is widely distributed in Southeast Asia and South Asia [4,6]. Thais have historically utilized the plant to treat fever, foodborne illness, gastroenteritis, aphthous ulcers, and cancer, as well as using it as an antiseptic agent [6]. The traditional use of this plant is based on the fact that its leaves are high in vitamin C and phenolic compounds. A discussion on the ethnopharmacology, phytochemistry, and botany of the *S. androgynus* plant had been presented [4].

On the other hand, there is evidence to suggest that the proximate makeup of the minerals and vitamins found in *S. androgynus* is almost certainly not the same as what was discovered in India and south China [4]. Environmental stresses may be triggered to contribute significantly to the differences found in the relative composition of the multiplicity of phytometabolites and biosynthesized in herbs; therefore, the origin is an important factor determining the element content and even quality [7]. These findings support the notion that cultivating sites with varying conditions may also impact phytochemical profiles.

There are few studies on the chemical composition and biological features of leaves [8,9], and to our knowledge, there are few data on the effects of *S. androgynus* leaves on ruminant animal models [10]. The objectives of this study were to investigate the phenolic profile of *S. androgynus* leaf extracts using high-performance liquid chromatography with diode array detection (HPLC-DAD) and to assess their biological potential for ruminant health. The antioxidant activity was tested against the radicals DPPH, nitric oxide (^•^NO), and superoxide (O_2_^•−^). *S. androgynus* leaves were also tested for their α-glucosidase inhibitory potential and protection against induced oxidative damage in ruminant erythrocytes. This is, as far as we know, the first study on the ability of *S. androgynus* leaves to inhibit the α-glucosidase enzyme, scavenge ^•^NO and O_2_^•−^ species, and protect ruminant erythrocytes against peroxyl radicals (ROO^•^) with regard to inhibition of hemoglobin oxidation, lipid peroxidation, and hemolysis. Furthermore, because nutritional antioxidants may be related to some of the purported positive benefits of the crude herb, this study was performed to isolate, characterize, and quantify the chemical composition of two methanolic and hexanoic extracts derived from extracts from each commercially and noncommercially cultivated site.

## 2. Results

### 2.1. Phytochemistry Profiling

Phytochemical components were effectively extracted from *S. androgynus* leaves using methanol or hexane without the need for time-consuming fractionation. Using HPLC-DAD, 14 phytochemical components were identified: vitamins (ascorbic acid), essential oils (eugenol), tannins (gallic acid), cinnamic acids (caffeic acid, syringic acid, *p*-coumaric acid, sinapic acid, and ferulic acid), and flavonoids (catechin, rutin, myricetin, quercetin, apigenin, and kaempferol). However, the clear peak and strong resolution of these phytochemical constituents were found in the methanolic plant extract 39 min after the start of the chromatographic run (Figure 1). The vitamin, essential oil, tannin, cinnamic acid, and flavonoid content of *S. androgynus* leaves could be recovered by methanolic plant extracts, but not myricetin. However, hexanoic plant extracts could compensate for their lack of myricetin by containing around 0.18% of total flavonoids. The differences between methanolic and hexanoic plant extracts were most significant for ascorbic acid and rutin, which were 11.44 and 2.24 times greater in methanolic versus hexanoic plant extracts, respectively (Table 1). Table 2 shows the total tannin, flavonoid, cinnamic acid, essential oil, and vitamin concentrations of commercial and noncommercial *S. androgynus* leaves collected for five consecutive months. The overall tannin, flavonoid, cinnamic acid, essential oil, and vitamin levels in the leaves of *S. androgynus* were unaffected by cultivar location or sample period.

### 2.2. Antioxidant Activity

Antioxidant activity of two methanolic and hexanoic plant extracts were determined in a concentration-dependent manner in the scavenging system of the radicals DPPH^•^, ^•^NO, and O_2_^•−^ (Figure 2A–C, respectively). The IC_50_ values of those plant extracts are summarized in Table 3. Compared to hexanoic plant extracts, methanolic plant extracts shown a greater capacity to scavenge DPPH^•^, ^•^NO, and O_2_^•−^ (*p* < 0.05).

### 2.3. α-Glucosidase Inhibitory Activity

The α-glucosidase inhibitory potential of two methanolic and hexanoic plant extracts was determined in a concentration-dependent manner (Figure 2D). The methanolic extract had a minimum in vitro α-glucosidase inhibitory potential of 37.04% at 10 mg concentration, and a maximum in vitro activity of 81.03% at 500 mg. The minimal in vitro α-glucosidase inhibitory potential of the hexanoic extract was 9.26% at a concentration of 10 mg, while its highest in vitro activity was 72.92% at 500 mg. The IC_50_ values of those plant extracts are presented in Table 3. The α-glucosidase inhibitory potential of methanolic plant extracts is stronger than that of hexanoic plant extracts (*p* < 0.05).

### 2.4. Evaluation of Leaf Extracts on Oxidative Damage in Ruminant Blood Erythrocytes

The in vitro ROO^•^-induced oxidative damage evaluation in ruminant erythrocytes of two methanolic and hexanoic plant extracts was assessed concentration-dependently for inhibition of hemoglobin oxidation, lipid peroxidation, and hemolysis (Figure 3). In Table 3, the IC_50_ values of these plant extracts are summarized. In comparison to hexanoic plant extracts, methanolic plant extracts inhibited hemoglobin oxidation, lipid peroxidation, and hemolysis more effectively (*p* < 0.05).

## 3. Discussion

### 3.1. Phytochemistry Profiles of Leaf Extracts Depended on Solvent Used, but Not Cultivation Site or Time Sampling

Soxhlet extraction is one of the sophisticated methods utilized to examine the chemical composition of medicinal plants. The extraction yield in reversed-phase liquid chromatography was discovered to be altered not only by the extraction procedure but also by the solvent used [8,11]. This is congruent with the current findings, which reveal that when compared to hexanoic plant extracts, methanolic plant extracts increase the resolution of the separated peak area and the consistency of the observed retention length. One potential rationale is that methanol may play an essential role in the recovery of phytochemicals from plant matrices. Methanol, which has a lower boiling point than hexane, is frequently used as an organic eluent in the presence of polar and nonpolar molecules. Thus, the lower boiling point and broad polarity of methanol may be potent attractants for the specific bioactive molecules of *S. androgynus* leaves, allowing for easier elution of the phytochemicals to methanol than to hexane [8]. Despite being less prevalent than methanol, the use of hexane in the current investigation could explain the successful recovery of complex plant matrices of *S. androgynus* from both polar and nonpolar substances. To determine phytochemicals from *S. androgynus*, however, the most recent review only included the use of organic solvents without mentioning the use of hexane [12]. Consequently, the current study may verify earlier research and provide additional evidence to support the concept of using alternative organic solvents for *S. androgynus*.

It is widely known that the vitamin content of *S. androgynus* is higher than that of most commonly used vegetables, demonstrating that *S. androgynus* is a multivitamin plant [4]. Similarly, we discovered that vitamin C or ascorbic acid has the highest concentration of phytochemicals in *S. androgynus* leaves. Indeed, the amount of vitamin C found in later observations of *S. androgynus* leaves appears to be lower than that found in Indian [13] or Chinese [14] *S. androgynus* leaves.

Our findings that *S. androgynus* leaves contain an abundance of flavonoids are consistent with prior research [15,16,17]. In the current study, flavonoids in *S. androgynus* leaves were ranked in the following order: rutin > apigenin > quercetin > catechin > kaempferol > myricetin. Overall, it appears that *S. androgynus* leaves in the current study contain more flavonoids than *S. androgynus* leaves from Indonesia [17] or India [16].

To the best of the author’s knowledge, no information exists on the tannin, essential oil, and cinnamic acid profiles of *S. androgynus* leaves. Previous studies merely established the presence of tannin [18] and essential oil [19] in *S. androgynus* leaves, but no estimate of their amounts was provided. We determined via HPLC that *S. androgynus* leaves contain at least 0.04 mg/g gallic acid, 0.32 mg/g eugenol, 0.01 mg/g caffeic acid, 0.005 mg/g syringic acid, 0.05 mg/g *p*-coumaric acid, 0.08 mg/g sinapic acid, and 0.04 mg/g ferulic acid. Therefore, the present findings may corroborate prior observations and suggest that *S. androgynus* leaves may be better capable of regulating a variety of biological processes.

There is evidence to suggest that the phytochemicals discovered in *S. androgynus* are almost certainly not the same as what was discovered in a number of Asian regions [4]. Despite differences in growth places and sample times, our data revealed that the overall levels of tannins, flavonoids, cinnamic acids, essential oils, and vitamins in commercial *S. androgynus* (CSA) and noncommercial *S. androgynus* (NSA) leaves were similar. This might be due to the absence of specific interactions between soil processes and nitrogen cycling in the Thai *S. androgynus* examined here [20]. No significant change in the environment could be attributed to the similarities found in the relative composition of the many phytometabolites and biosynthesized in the leaves of the *S. androgynus* plant. In the same way, our earlier studies [8,11] found similarities in the phytochemical profiles of Thai medicinal herbs, even though the plants came from different places and were sampled at different times.

### 3.2. Antioxidant Activity

Normal cellular metabolism produces reactive oxygen species such as superoxide anion, nitric oxide radical, and (peroxyl) ROO^•^. The superoxide anion and the nitric oxide radical are two of these radicals that are known for playing significant roles in the emergence of numerous pathophysiologic conditions. When superoxide reacts with certain transition metal ions, it produces a highly reactive oxidizing species known as the hydroxyl radical. The presence of these radicals within the ruminant body had resulted in the development of endogenous systems that are able to reduce the quantities of these radicals [2,21]. On the other hand, when these defense systems did not work or were not enough, the antioxidants that are present in the diet become extremely important as potential protective agents [22,23,24].

Numerous tests, such as the DPPH^•^, superoxide, and nitric oxide assays, can determine the antioxidant capacity of bioactive compounds and plant extracts. These tests give a screening of the extracts’ and compounds’ antioxidant potential [25]. The antioxidant activity of methanolic extracts of *S. androgynus* leaf tested against DPPH^•^, superoxide anion, and nitric oxide radicals in the current investigation was comparable to previous data [4,26].

However, as far as we know, this is the first study to look at the antioxidant capacity of hexanoic *S. androgynus* leaf extract. When compared to hexanoic extracts, the superior results of the methanolic extract of *S. androgynus* leaves demonstrated a tendency to be more potent against DPPH and nitric oxide radicals. This notion is consistent with the vitamin and flavonoid makeup of leaves, where ascorbic acid and rutin were detected in much higher concentrations in methanolic plant extracts. It is well documented that ascorbic acid and rutin are among the most powerful free radical scavengers [27].

Although the existence of other phytochemical substances not detected in the extracts cannot be ruled out, the presence of higher levels of tannin, essential oils, and cinnamic acids in methanolic plant extracts may be linked to their increased antioxidant properties. When the hexanoic assay findings were compared, we found that the methanolic extracts were more potent against superoxide radicals. The removal of these reactive oxygen species is crucial due to their propensity to generate other reactive species that are particularly damaging to cells.

### 3.3. Evaluation of α-Glucosidase Inhibitory Activity

To date, *S. androgynus* leaves had been shown to significantly lower blood sugar levels in human patients [28]. However, no dose–effect data are available, and the particular biologically active constituent (s) in these leaves responsible for hypoglycemic activity are unknown [4]. This is the first study to evaluate the physiologically active constituent (s) and their derivatives in *S. androgynus* leaves in relation to ruminant glucosidase inhibition. In this study, it was discovered that methanolic plant extract inhibits α-glucosidase more effectively than hexanoic plant extract because it contains more cinnamic acids and flavonoids. Our findings suggested that the presence of a hydroxyl group in cinnamic acid is critical for inducing significant inhibition of α-glucosidase. Cinnamic acids, particularly caffeic acid and ferulic acid, are widely thought to block intestinal α-glucosidase, which is one of the treatment techniques [29]. Previous research showed that ferulic acid or caffeic acid directly enhanced pancreatic insulin secretion, resulting in lower plasma glucose concentrations in humans [30]. In addition, ferulic acid (50 mg/kg) reduced blood glucose levels while increasing plasma insulin levels in type 2 diabetic mice by increasing hepatic glycogen production and glucokinase activity [30]. Furthermore, the flavonoid structure, position, and number of hydroxyl groups all influence the desired impact [31]. Among flavonoids, for example, quercetin is a more potent inhibitor. That said, quercetin may be responsible for the greater inhibitory impact, which appears to increase the interaction between α-glucosidase and flavonoids, whereas the meta position of hydroxyl groups reduced the electron cloud density, resulting in a lesser inhibitory activity. Based on the results of previous published studies using cinnamic acids or flavonoids derived from plants, it is speculated that cinnamic acid or flavonoid derivatives in *S. androgynus* leaves may contribute to a mechanism in the control of hyperglycemia by inhibiting α-glucosidase, resulting in a decrease in hemoglobin A1c (HbA1c). HbA1c reduction may lower the occurrence of chronic vascular problems in diabetic ruminants. Unfortunately, neither pure nor fragmented cinnamic acid or flavonoids from *S. androgynus* leaves were assessed separately in the current in vitro investigation; thus, additional research is needed to shed light on this issue.

### 3.4. Protective Effects of Leaf Extracts on Oxidative Damage in Ruminant Blood Erythrocytes

Erythrocytes are the most common type of circulating cell in mammals, and their primary function is gas exchange during respiration. Furthermore, these cells participate in immunologically complex reactions (antibodies, complements, and bacteria [32]). Their membranes, however, are rich in polyunsaturated fatty acids. This, combined with the fact that they transport oxygen, makes them ideal targets for free radicals and potential promoters of reactive oxygen species (ROS [25,33]). In the presence of oxygen, the formation of alkyl radicals can result in hemoglobin oxidation, lipid peroxidation, and finally, hemolysis [25]. Recent research has linked the long-term accumulation of reactive oxygen species caused by a high temperature–humidity index to decreased ruminant animal productivity [34,35] and immunological function in tropical environments [1,2,23].

We tested the ability of methanolic and hexanoic leaf extracts to prevent ROO^•^ production, the reactive oxygen species responsible for triggering hemoglobin oxidation and producing methemoglobin. Here, we present the initial report of a *S. androgynus* leaf-based inhibitory assay for hemoglobin oxidation.

The synthesis of methemoglobin results from the oxidation of hemoglobin when the iron in the heme group is not in its normal state. This leads to the production of oxidative stress events, the destruction of lipids and the modification of protein interactions, which in turn affect the equilibrium and resistance of the erythrocytes’ membrane [36]. The TBARS test is the most commonly used method for determining how much lipid peroxidation has occurred in biomaterials [2,25]. This is the first study of its kind to examine whether leaves can inhibit lipid peroxidation in ruminant erythrocytes.

Consistent with previous studies [37,38,39], our study indicated that ascorbic acid, eugenol, gallic acid, cinnamic acids (caffeic acid, syringic acid, *p*-coumaric acid, sinapic acid, and ferulic acid), and flavonoids (catechin, rutin, myricetin, quercetin, apigenin, and kaempferol) and their derivatives have hydroxyl group substitutions that are associated with their protective potential. Flavonoids, for instance, had been demonstrated to increase antihemolytic protection due to their chemical structure and well-known liposolubility (the C2=C3 link of flavonoids’ C ring increased antioxidant capacity [40]). This allowed them to absorb into the membrane and function as antioxidants, preventing membrane damage by removing dangerous species before they could cause damage. Indeed, the phytochemical makeup of *S. androgynus* leaves was detected in much higher concentrations in methanolic plant extracts and could be expected to have more hydroxyl groups acting as antioxidant agents in mammalian circulating cells.

### 3.5. Limitations of The Study

As assessed only in ruminant erythrocytes, our findings demonstrated a moderate improvement in the potential for phytochemicals from *S. androgynus* leaves to serve as a viable alternative to natural antioxidants. Empirical investigation of other liquid blood, such as plasma, as well as gene expression and oxidative indicators in ruminal fluids, could help to support this notion. This could be attributed in part to the limitations of our investigation. Another limitation is the lack of further measurements to back up the results of the rumen fermentation data. Indeed, rumen microbial habitats can be altered by flavonoids, tannins, or essential oils, with implications for ruminant lifetime performance and welfare. This is consistent with studies that found substantial impacts in lowered oxidative stress indicators in physiological fluids (e.g., blood, ruminal fluid, and milk) and physiological tissues (e.g., mammary gland) of early-lactation goats fed *Piper betle* leaves, which contain flavonoids, essential oils, and phenolic acids [2,41]. These goats also underwent positive nutritional digestion and fermentation.

## 4. Materials and Methods

### 4.1. Standards and Reagents

HPLC-grade solvents, including glacial acetic acid, acetonitrile, methanol, and hexane used in the extraction process, were purchased from Labscan (Bangkok, Thailand); the purities of the solvents were above 99%. Acarbose, tannins (gallic acid), flavonoids (catechin, rutin, myricetin, quercetin, apigenin and kaempferol), cinnamic acids (caffeic, syringic, *p*-coumaric, ferulic, and sinapic acid), and essential oils (eugenol) were purchased from Sigma Chemical (purity: >99%, St. Louis, MO, USA). Vitamin C (ascorbic acid), used as a vitamin standard, was purchased from Carlo Erba (purity: >99%, Strada Rivoltana, France). Water for preparation, extraction, and liquid chromatography was prepared using a Milli-Q water purification system (Millipore, Illkirch-Graffenstaden, France). 1,1-Diphenyl-2-picrylhydrazyl (DPPH^•^), β-nicotinamide adenine dinucleotide (NADH), phenazine methosulfate (PMS), nitrotetrazolium blue chloride (NBT), α-glucosidase from *Saccharomyces cerevisiae* (type I, lyophilized powder), phosphate-buffered saline (PBS), trypan blue and 2,2′-azobis (2-ethylpropionamidine) dihydrochloride (AAPH), thiobarbituric acid (TBA), trichloroacetic acid (TCA), and tert-butyl hydroperoxide (t-BHP) were purchased from Sigma-Aldrich (St. Louis, MO, USA). *N*-(1-naphthyl) ethylenediamine dihydrochloride, sulfanilamide, 4-nitrophenyl-alpha-D-glucopyranoside (pNPG), and sodium nitroprusside dihydrate (SNP) were obtained from Merck KGaA (Darmstadt, Germany).

### 4.2. Leaf Sample, Extraction, and Phytochemistry Profiling

Following previous methods [8,42], the procedure for gathering leafy materials was carried out. Plant samples were separated into two groups: commercial *S. androgynus* (CSA) and noncommercial *S. androgynus* (NSA) leaves. Thailand’s northern (Chiang Rai), western (Phetchaburi), eastern (Prachinburi), and southern (Songkhla) local marketplaces were where the CSA were acquired. NSA was grown at two different locations: the Suranaree University of Technology Organic Farm (14°87′1593″ N, 102°02′5890″ E) and the temporary garden near the Goat and Sheep Research Center (14°88′0532″ N, 102°00′4633″ E), both in Nakhon Ratchasima (Northeastern, at an elevation of 243 m above sea level). CSA and NSA samples were collected for five consecutive months from fresh sources (April–August 2019). Following flower separation (if any), the leaves were rinsed, blanched with steam at 90 °C for 1 min, and stored at 20 °C until further investigation.

Freeze-drying of CSA and NSA samples was conducted using a Lyophilizer (GAMMA 2–16 LSC, Christ, Osterode am Harz, Germany). For 24 h, the plant samples were frozen at −80 °C and dried under a vacuum with the condenser temperature set at −15 °C. The lyophilized leaves were homogenized and powdered in a Retsch mill with a mesh size of 1 mm (Retsch SM 100 mill, Haan, Germany). Powdered leaves were put in plastic bags and kept in a vacuum desiccator at 25 °C and 34% humidity until they were used.

Vitamins (ascorbic acid), essential oils (eugenol), tannins (gallic acid), cinnamic acids (caffeic acid, syringic acid, *p*-coumaric acid, sinapic acid, and ferulic acid), and flavonoids (catechin, rutin, myricetin, quercetin, apigenin, and kaempferol) evaluated in methanol and hexane were extracted utilizing a Soxhlet extraction system, as described previously [8,11]. In brief, 5 g of dried leaf powder was extracted with 20 mL hexane using the Soxhlet equipment for 3–4 h, and then plant extracts were obtained. These Soxhlet techniques were performed three times and all plant extracts were mixed. The recovered plant extract was then evaporated using a Rotavapor (Buchi R300, Flawil, Switzerland). The plant extract was filtered using a 0.45 m Polyvinylidene difluoride (PVDF) syringe filter (Merck, Darmstadt, Germany) and decanted into a volumetric flask. The volume of recovered plant extract was increased to 10 mL using the appropriate solvents and stored at 20 °C for further liquid chromatography analysis. The plant extract extracted with methanol followed the same techniques as the hexane extract.

The gathered plant extracts were evaporated on the working day to obtain a dry matter [43,44,45]. One milligram of evaporated plant extract was dissolved in 0.5 mL of mobile-phase solution (1:9, HPLC-grade acetonitrile: 1% acetic acid), vortexed, and filtered through a 0.45 µm Polyvinylidene difluoride (PVDF) syringe filter. The preparation of the standard stock solution, calibration standard, quality control sample, chromatographic conditions, and computation were carried out in accordance with a prior procedure [8,46]. In brief, twenty injections of each prepared extract (methanolic or hexanoic) from leaves over the course of five consecutive months (*n* = 20), with four representative replications from each month, were equipped and analyzed on an HPLC system (Agilent Technologies1260 Infinity, Santa Clara, CA, USA) for 65 min using a reversed-phase Zorbax SB-C18 column (3.5 µm particle size, i.d. 4.6 mm × 250 mm, Agilent Technologies, Santa Clara, CA, USA). The HPLC setup consisted of four quaternary pumps for the solvent delivery system (61311 B), a DAD (61315 D), a 10 mm flow cell, and an automated sample injection valve with a 100 µL loop. An Agilent OpenLAB CDS 1.8.1 system manager was used to accomplish data integrity and chromatographic data analysis. The mobile phase contained 1% acetic acid and HPLC grade acetonitrile (1:9). To achieve chromatographic separation, a flow rate of 0.9 mL/min was used with a binary gradient of (A) acetonitrile and (B) 1% acetic acid. The gradient elution system consisted of the following levels: 10–40% A (0–28 min), 40–60% A (28–39 min), 60–90% A (39–50 min), and 90–10% A (50 min) (50–65 min). The absorption of the substances that were tested and measured was evaluated using a photodiode array UV detector set at 272 nm. Within 65 min, 14 external standards of tannins, flavonoids, cinnamic acids, essential oils, and ascorbic acid were effectively manufactured, mixed, and analyzed. The peak area was employed for quantification (dilution included), with a respective external standard calibration curve used in accordance with the previous approach. Values were given as mg/g on a dry weight basis (Table 1).

### 4.3. Antioxidant Activity

The antioxidant activity of methanolic and hexanoic extracts of *S. androgynus* leaves against the radicals DPPH^•^, nitric oxide (^•^NO), and superoxide (O_2_^•−^), was measured using a previously established method [47]. Ascorbic acid was employed as a positive control. Each experiment was repeated six times and the results were expressed as IC_50_ values (mg/mL).

For DPPH^•^ assay, all components of the samples were previously redissolved in methanol (25 μL) and placed in the different wells of the microplate, followed by the addition of 200 μL of 150 mM methanolic DPPH. For each extract, 16 different dilutions were prepared, placed into a 96-well plate, and read at 515 nm.

For O_2_^•−^ assay, all components of the samples were dissolved in phosphate buffer (19 mM, pH 7.4). For each extract, 16 different dilutions were prepared, placed into a 96-well plate, tested using NBT, and monitored at 562 nm.

For ^•^NO assay, all components of the samples were dissolved in phosphate buffer (100 mM, pH 7.4). For each extract, 16 different dilutions were prepared and placed into a 96-well plate. The chromophore formed with Griess reagent was read at 562 nm.

### 4.4. α-Glucosidase Inhibitory Activity

The inhibition of α-glucosidase activity was assessed using the Ellman’s method, as previously described [48]. Acarbose was employed as a positive control. Each experiment was repeated six times. Sixteen different concentrations were tested using a 96-well-plate. Each well contained 50 μL of the *S. androgynus* leaf extract dissolved in potassium phosphate buffer, 150 μL of potassium phosphate buffer, and 100 μL of 4-nitrophenyl-α-D-glucopyranoside (PNP-G). The reaction was initiated by the addition of 25 μL of the enzyme, and after incubation, the absorbance was measured at 405 nm.

### 4.5. In Vitro ROO^•^-Induced Oxidative Damage in Ruminant Erythrocytes

To evaluate the in vitro ROO^•^-induced oxidative damage in ruminant erythrocytes in terms of inhibition of hemoglobin oxidation, lipid peroxidation, and hemolysis, one milligram of evaporated plant extract (methanolic or hexanoic) was dissolved in one milliliter of PBS, and sixteen different dilutions were made. As a positive control, quercetin was used. Each experiment was carried out six times, with the findings given as IC_50_ values (mg/mL).

Blood samples (5 mL) were collected from the jugular veins of random dairy goats at the university farm and deposited in evacuated tubes containing K_3_EDTA. Feeding management and animal ethics considerations for used dairy goats had been reported [49,50,51]. Erythrocytes were isolated using the method described before [21,49].

The inhibition of hemoglobin (Hb) oxidation was measured by the ability of *S. androgynus* extracts to reduce methemoglobin production following to previous report [25]. Methemoglobin is produced when oxyhemoglobin combines with AAPH at a temperature of 38 °C in a water bath. This results in the disintegration of AAPH, which is dissolved in PBS. The reaction mixture was formed by combining 100 μL of previously prepared PBS-fixed extract with 200 μL of erythrocyte solution (1250 × 10^6^ cells/mL, final density). The control and blank were conducted by substituting 100 μL of PBS for the sample. The reaction mixtures were incubated for 30 min in a water bath at 38 °C with slow agitation (50 rpm). Following incubation, 200 μL of AAPH (50 mM final concentration) was added to the mixture (except in the blank), followed by 4 h of incubation under the identical conditions described previously. The entire volume (500 μL) was transferred to 1.5 mL conical Eppendorf tubes and centrifuged for 6 min at 4 °C at 1500× *g*. The supernatant (300 μL) was deposited in a 96-well plate, and the absorbance at 630 nm was measured.

The production of thiobarbituric acid-reactive compounds was used to indirectly measure lipid peroxidation in erythrocytes (TBARS) following the previous report [25]. Previous PBS-fixed extract was combined with a suspension of goat cells (500 × 10^6^ cells/mL, final density) at 38 °C for 30 min with mild agitation (50 rpm). Following incubation, *tert*-butyl hydroperoxide (tBHP, 0.2 mM final concentration) was added to the medium, which was then incubated for 30 min at 38 °C with mild agitation. After incubation, the entire contents were collected and transferred to a 1.5 mL conical Eppendorf tube, and 28% (w/v) trichloroacetic acid (TCA) was added to induce protein precipitation, followed by a 10 min centrifugation at 16,000× *g* at 18 °C. To produce TBARS from malondialdehyde (MDA) and thiobarbituric acid (TBA), the supernatant was placed in a 2 mL conical test tube (with screw cover), followed by the addition of TBA 1% (w/v). The resulting mixture was then placed in a water bath and subjected to heating for 15 min at 38 °C. The test tubes were then cooled to room temperature, and the absorbance at 532 nm was measured.

ROO^•^ generation and lysis events are caused by AAPH thermal breakdown. The hemolysis of goat erythrocytes was assessed by measuring Hb release after membrane breakage caused by the hemolytic process, as detailed in the optimal protocol [25]. Previous PBS-fixed extract (100 μL) was mixed with 200 μL of goat erythrocyte suspension (1775 × 10^6^ cells/mL) and incubated in a water bath at 38 °C for 30 min with mild agitation (50 rpm). Following the incubation period, 200 μL of AAPH (17 mM, final concentration) was added to the mixture (except in the blank), followed by a 3 h incubation period under the same conditions. The entire volume (500 μL) was transferred to a 1.5 mL conical Eppendorf tube and centrifuged for 5 min at 4 °C at 1500× *g*. The supernatant (300 μL) was transferred to a 96-well plate, and absorbance at 540 nm was measured.

### 4.6. Statistical Analysis

Phytochemistry profiling on total vitamin, essential oil, tannin, cinnamic acid and flavonoid concentrations in leafy extracts cultivated in two locations and sampled at five distinct periods was calculated using GraphPad Prism 9.0 and a completely randomized design with repeated measures (GraphPad Software, Inc., San Diego, CA, USA). The mixed-effects model of GraphPad Prism’s Akaike’s information criterion was used to fit the covariance structure of the compound symmetry. The Kolmogorov–Smirnov test was used to ensure that the data were normally distributed. The Student’s test was used to evaluate data on antioxidant, antidiabetic, and antihemolytic properties. Means calculated using least squares were reported, and significance was determined using Tukey’s honestly significant difference (HSD) at a level of *p* < 0.05.

## 5. Conclusions

Our research found vitamins (ascorbic acid), essential oils (eugenol), tannins (gallic acid), cinnamic acids (caffeic acid, syringic acid, *p*-coumaric acid, sinapic acid, and ferulic acid), and flavonoids in *S. androgynus* leaves (catechin, rutin, myricetin, quercetin, apigenin, and kaempferol). The differences in those phytochemistry profiles were determined by the solvent type, cultivated site, and time sampling. Surprisingly, the physicochemical profiles of leaf extracts were affected by the solvent used, but not by the location of the plant or the time of sampling. Our findings also supported the hypothesis that the type of solvent used inhibited the activities of DPPH, nitric oxide, superoxide, and the α-glucosidase enzyme, as well as modulated the inhibition of ruminant erythrocyte hemoglobin oxidation, lipid peroxidation, and hemolysis. These advantages could be attributed to *S. androgynus* leaves, a potential source of phytochemical compounds with health-promoting properties for ruminant production. These findings, if supported by at least in vitro studies on nutritional digestion and fermentation, would have the potential to open up new avenues in the search for natural antioxidants and antimicrobials that could be successfully tested in future in vivo studies.

## Figures and Tables

**Figure 1 molecules-27-08580-f001:**
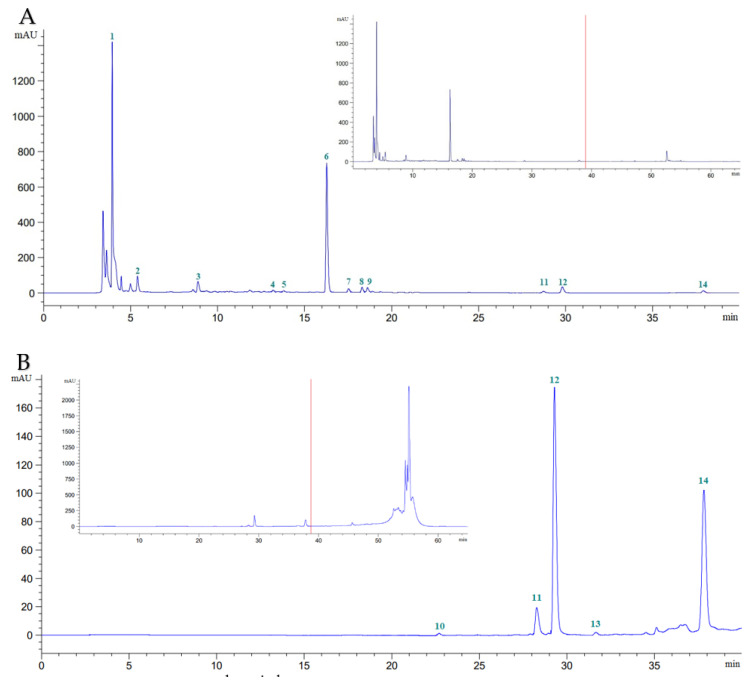
Chromatogram of *S. androgynus* leaves at λ = 272 nm, flow rate 0.9 mL/min, for 65 min. (**A**), methanolic extract (20 × dilution); (**B**), hexanoic extract (20 × dilution). Peak 1, ascorbic acid; 2, gallic acid; 3, catechin; 4, caffeic acid; 5, syringic acid; 6, rutin; 7, *p*-coumaric acid; 8, sinapic acid; 9, ferulic acid; 10, myricetin; 11, quercetin; 12, apigenin; 13, kaempferol; 14, eugenol.

**Figure 2 molecules-27-08580-f002:**
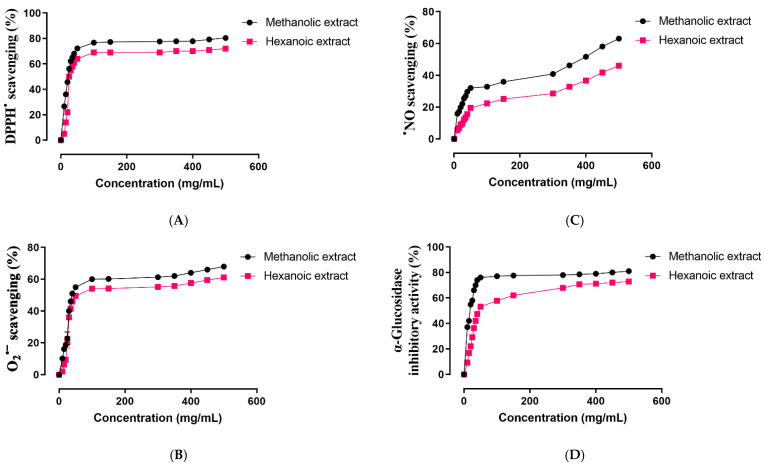
Effect of methanolic and hexanoic extracts of *S. androgynus* leaves against (**A**), DPPH^•^: 1,1-diphenyl-2-picrylhydrazyl; (**B**), O_2_^•−^: superoxide; (**C**), ^•^NO: nitric oxide; (**D**), α-glucosidase.

**Figure 3 molecules-27-08580-f003:**
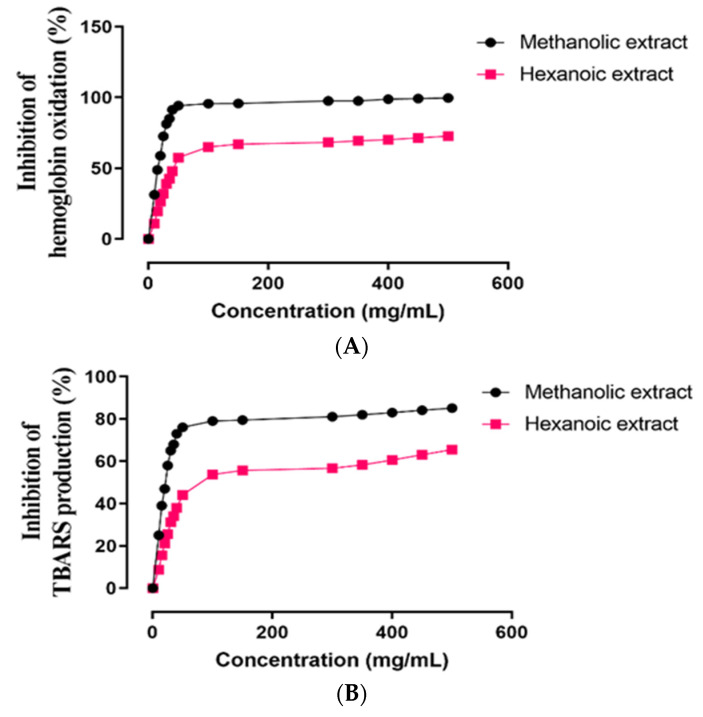
Effect of methanolic and hexanoic extracts of *S. androgynus* leaves against (**A**) hemoglobin oxidation; (**B**) lipid peroxidation; (**C**) hemolysis.

**Table 1 molecules-27-08580-t001:** Identifying and quantifying selected phytochemicals in methanolic and hexanoic extracts of commercial and noncommercial *S. androgynus* leaves.

Organic Compound	Wavelength Detection (nm)	Concentration (mg/g on Dry Weight Basis) ^a^		
		Methanol	Hexane	Average
CSA				
Ascorbic acid	272, 280, 310	11.45 ± 0.12	nd	5.72 ± 0.06
Gallic acid	272, 280, 310	0.07 ± 0.01	nd	0.03 ± 0.003
Catechin	272, 280, 310	0.13 ± 0.03	nd	0.06 ± 0.02
Caffeic acid	272, 280, 310	0.02 ± 0.04	nd	0.01 ± 0.002
Syringic acid	272, 280, 310	0.01 ± 0.002	nd	0.004 ± 0.001
Rutin	272, 280, 310	2.23 ± 0.18	nd	1.12 ± 0.09
*p*-coumaric acid	272, 280, 310	0.07 ± 0.01	nd	0.03 ± 0.01
Sinapic acid	272, 280, 310	0.17 ± 0.03	nd	0.08 ± 0.02
Ferulic acid	272, 280, 310	0.07 ± 0.02	nd	0.03 ± 0.01
Myricetin	272, 280, 310	nd	0.004 ± 0.0004	0.002 ± 0.0002
Quercetin	272, 280, 310	0.15 ± 0.03	0.02 ± 0.002	0.09 ± 0.02
Apigenin	272, 280, 310	nd	1.82 ± 0.05	0.91 ± 0.02
Kaempferol	272, 280, 310	nd	0.01 ± 0.0003	0.005 ± 0.0002
Eugenol	272, 280, 310	0.15 ± 0.04	0.40 ± 0.01	0.27 ± 0.02
NSA				
Ascorbic acid	272, 280, 310	11.43 ± 0.20	nd	5.71 ± 0.10
Gallic acid	272, 280, 310	0.06 ± 0.01	nd	0.03 ± 0.005
Catechin	272, 280, 310	0.12 ± 0.03	nd	0.06 ± 0.02
Caffeic acid	272, 280, 310	0.02 ± 0.004	nd	0.01 ± 0.002
Syringic acid	272, 280, 310	0.01 ± 0.004	nd	0.003 ± 0.002
Rutin	272, 280, 310	2.25 ± 0.09	nd	1.12 ± 0.04
*p*-coumaric acid	272, 280, 310	0.07 ± 0.01	nd	0.03 ± 0.01
Sinapic acid	272, 280, 310	0.16 ± 0.01	nd	0.08 ± 0.003
Ferulic acid	272, 280, 310	0.07 ± 0.02	nd	0.03 ± 0.01
Myricetin	272, 280, 310	nd	0.004 ± 0.001	0.002 ± 0.001
Quercetin	272, 280, 310	0.15 ± 0.04	0.03 ± 0.005	0.09 ± 0.02
Apigenin	272, 280, 310	nd	1.80 ± 0.20	0.90 ± 0.10
Kaempferol	272, 280, 310	nd	0.01 ± 0.002	0.005 ± 0.001
Eugenol	272, 280, 310	0.13 ± 0.01	0.42 ± 0.12	0.28 ± 0.06

^a^ Data are reported as mean ± SD (*n* = 20 for each solvent), with four representative replications from each month. nd: not detectable. CSA: commercial *S. androgynus* leaves. NSA: noncommercial *S. androgynus* leaves.

**Table 2 molecules-27-08580-t002:** Total contents of tannins, flavonoids, cinnamic acids, essential oils, and vitamins of commercial and noncommercial *S. androgynus* leaves (mg/g on dry weight basis).

Organic Compound	CSA	NSA	SEM	*p*-Value ^1^		
				Cultivated Site	Sampling Time	Interaction
Ascorbic acid	5.77	5.78	0.026	0.601	0.994	0.573
Gallic acid	0.04	0.04	0.007	0.590	0.927	0.328
Catechin	0.08	0.08	0.012	0.969	0.420	0.961
Caffeic acid	0.01	0.01	0.003	0.352	0.925	0.684
Syringic acid	0.005	0.005	0.001	0.510	0.467	0.304
Rutin	1.14	1.19	0.035	0.115	0.798	0.841
*p*-coumaric acid	0.05	0.05	0.010	0.776	0.526	0.438
Sinapic acid	0.08	0.08	0.008	0.337	0.890	0.059
Ferulic acid	0.05	0.04	0.010	0.787	0.588	0.345
Myricetin	0.005	0.004	0.001	0.549	0.350	0.737
Quercetin	0.11	0.09	0.017	0.201	0.762	0.349
Apigenin	0.99	0.99	0.058	0.869	0.997	0.720
Kaempferol	0.01	0.01	0.002	0.673	0.519	0.345
Eugenol	0.32	0.33	0.031	0.694	0.132	0.168
Total tannin	0.04	0.04	0.007	0.590	0.927	0.328
Total flavonoid	2.34	2.36	0.071	0.712	0.733	0.429
Total cinnamic acid	0.20	0.18	0.020	0.456	0.741	0.075
Total essential oil	0.32	0.33	0.031	0.694	0.132	0.168
Total vitamin	5.77	5.78	0.026	0.601	0.994	0.573

^1^*p* Value: effect of cultivated site (CSA versus NSA), effect of sampling time (April–August), and their interaction (cultivated site × sampling time); total tannin: sum of gallic acid; total flavonoid: sum of catechin, rutin, myricetin, quercetin, apigenin and kaempferol; total cinnamic acid: sum of caffeic acid, syringic acid, *p*-coumaric acid, sinapic acid and ferulic acid; total essential oil: sum of eugenol; total vitamin: sum of ascorbic acid; SEM: standard error of mean.

**Table 3 molecules-27-08580-t003:** IC_50_ (mg/mL) values found in the antioxidant activity, α-glucosidase, hemoglobin oxidation, lipid peroxidation, and hemolysis assays for *S. androgynus* leaves.

Item	Methanolic Extract	Hexanoic Extract
DPPH^•^	13.14 ± 0.055 ^a^	18.38 ± 0.061 ^b^
^•^NO	55.02 ± 1.338 ^a^	129.40 ± 3.114 ^b^
O_2_^•−^	25.31 ± 0.886 ^a^	27.48 ± 0.711 ^b^
α-Glucosidase	9.83 ± 0.032 ^a^	31.74 ± 0.020 ^b^
Hemoglobin oxidation	11.96 ± 0.011 ^a^	27.17 ± 0.207 ^b^
Lipid peroxidation	13.54 ± 0.012 ^a^	32.51 ± 0.027 ^b^
Hemolysis	5.940 ± 0.005 ^a^	18.07 ± 0.010 ^b^

Data are reported as mean ± SD (*n* = 6 for each assay); ^a,b^ values on the same row under each main effect with different superscript differ significantly (*p* < 0.05); DPPH^•^: 1,1-diphenyl-2-picrylhydrazyl; ^•^NO: nitric oxide; O_2_^•−^: superoxide.

## Data Availability

All data are contained within the article.

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
