# Peer review of "Exploring the Phytochemical Profiles and Antioxidant, Antidiabetic, and Antihemolytic Properties of Sauropus androgynus Dried Leaf Extracts for Ruminant Health and Production"

_molecules, 2022, doi:10.3390/molecules27238580_

Round 1

Reviewer 1 Report

This is a complete report.    

The attached file contains some comments.

Author Response

Dear reviewer#1

We provide a point-by-point response to the reviewer’s comments as a word file, please see the attachment.

Best regards

Reviewer 2 Report

Review Report

I would like to thank all authors of the manuscript for their good and novelty manuscript titled as (Exploring the Phytochemical Profiles, Antioxidant, Anti-Dia-2 betic, and Anti-Hemolytic Properties of Sauropus Androgynus Dried Leaf Extracts for Ruminant Health and Production) which submitted to journal (Molecules ).

1-The manuscript is original and novel as it aims to show the presence of vitamin (essential oil (eugenol), tannin (gallic 461 acid), cinnamic acids (caffeic acid, syringic acid, p-coumaric acid, sinapic acid, and ferulic 462 acid), and flavonoids in S. androgynus leaves and the differences in those phytochemistry profiles were determined by the solvent type, cultivated site, and time sampling. Surprisingly, the physicochemical profiles of leaf extracts were affected by the solvent used.

2-The Presentation of the manuscript is good which attract the Interest to the readers.

3-Minor revision is needed to English language and style.

4-The introduction provide sufficient background and include all relevant references and styled according to the style of the journal.

5- All the cited references relevant to the research.

6- All the cited references relevant to the research.

7- The methods adequately described.

8- The results clearly presented.

9- The conclusions supported by the results.

So, I recommend accepting after minor revision (corrections to minor methodological errors and text editing.

Corrections are:

In Titel

Corrections are highlighted in the page 1 line (2)

In Abstract

Corrections are highlighted in the page 1 lines (10,17,18,22,23,25,26,29) .

 In  Introduction

Corrections are highlighted in the page 2 lines(51,54,56.59,67,69,72,74,77,79)

In Results

Corrections are highlighted in the page 2,3,5,6

 lines  (84,88,91,92,97,98,100,102,107,111,112,114,115120,129,130,132,134,135,142,146)

In Discussion

Corrections are highlighted in the page 7,8 ,9,10

lines(161,164,165,167,169,173174,176,177,178,180,182,183,186,187,188,197,206,209,212,213,219,226,230,241,247,251,257,261,267,271,278,285,290)

In Material and method

Corrections are highlighted in the page10 ,11 ,12,13

lines(204,307,308,311,321,323,324,341,342,346,378,381,397,406,412,450)

Author Response

Dear reviewer#2

We provide a point-by-point response to the reviewer’s comments as a Word file, please see the attachment.

Best regards

Reviewer 3 Report

The authors report in detail the results of their research on dried leaf extracts of  Sauropus Androgynus, a shrub grown in tropical regions and used among other things as leaf vegetable and biomedical drugs. Especially the phytochemical profiles, the antioxidant, anti-diabetic and antihemolytic properties are describe and its use of the extracts for ruminant health and production. Their findings suggest that the leaves of S. androgynus are a prospective source of phytochemical substances with health-promoting qualities for ruminant production. Thus , their results could be of practical importance  for agrarians and the food industry.

The work is presented clearly and in good scientific  English. Slight improvements and corrections are needed before publication. Some hints are indicated directly in the attached review.  

A few points should be explained/added:

#162 and  #212-214: It is not clear WHY you use hexane as extraction solvent. For me this is rather unusual, because you are looking for chemical polar and medium polar subtances; and hexane is usually used for defatting and extraction of more or less unpolar subtances. Please explain.

Table 1 and 2: are not clearly explained.

Table 1: what are the SD values? Due to the explanation in the experimental part the n = 20 is simply a 20-times injection of the same extract solution and thus hint for the precision of your HPLC measurement, but not for the mean concentration of the substances in the plant. Please explain.

Table 2: What is SEM; How are the p-values calculated?

Author Response

Dear reviewer#3

We provide a point-by-point response to the reviewer’s comments as a Word file, please see the attachment.

Best regards
